# Implementation of Guidelines for Healthier Canteens in Dutch Secondary Schools: A Process Evaluation

**DOI:** 10.3390/ijerph16224509

**Published:** 2019-11-15

**Authors:** Irma J. Evenhuis, Ellis L. Vyth, Lydian Veldhuis, Suzanne M. Jacobs, Jacob C. Seidell, Carry M. Renders

**Affiliations:** 1Department of Health Sciences, Faculty of Science, Vrije Universiteit Amsterdam, Amsterdam Public Health research institute, 1081 HV Amsterdam, The Netherlands; info@ellisvyth.nl (E.L.V.); j.c.seidell@vu.nl (J.C.S.); carry.renders@vu.nl (C.M.R.); 2Netherlands Nutrition Centre, PO Box 85700, 2508 CK The Hague, The Netherlands; veldhuis@voedingscentrum.nl (L.V.); jacobs@voedingscentrum.nl (S.M.J.)

**Keywords:** process evaluation, implementation, school canteen, policy, nutrition

## Abstract

The Netherlands Nutrition Centre has developed ‘Guidelines for Healthier Canteens’. To facilitate their implementation, implementation tools were developed: stakeholders’ questionnaires, the ‘Canteen Scan’ (an online tool to assess product availability/accessibility), a tailored advisory meeting/report, communication materials, establishment of an online community, newsletters, and a fact sheet with students’ wishes/needs. In this quasi-experimental study, we investigated the effect of these tools in secondary schools on (a) factors perceived by stakeholders as affecting implementation; (b) the quality of implementation. For six months, ten intervention schools implemented the guidelines, supported by the developed implementation tools. Ten control schools received the guidelines without support. School managers, caterers, and canteen employees (*n* = 33) reported on individual and environmental factors affecting implementation. Implementation quality was determined by dose delivered, dose received, and satisfaction. Stakeholders (*n* = 24) in intervention schools scored higher on the determinants’ knowledge and motivation and lower on need for support (*p* < 0.05). Dose received (received and read) and satisfaction was highest for the advisory meeting/report (67.9%, 64.3%, 4.17), communication materials (60.7%, 50.0%, 3.98), and fact sheet (80%, 60%, 4.31). Qualitative analyses confirmed these quantitative results. In conclusion, a combination of implementation tools that includes students’ wishes, tailored information/feedback, reminders and examples of healthier products/accessibility supports stakeholders in creating a healthier school canteen.

## 1. Introduction

School is a useful setting in which to stimulate healthy dietary behaviour in adolescents [1,2]. National or regional policy focused on provision of healthier foods and drinks in canteens and vending machines in schools seems to encourage adolescents to eat more healthily during school time [3,4,5]. In the Netherlands, students bring most foods and drinks from home, as schools do not provide meals in the absence of a national/regional school meal plan. Most schools have a canteen and/or vending machines, where students buy substitutional snacks or drinks. Due to the absence of national guidance and international consensus on how to define a ‘healthier canteen’, the Netherlands Nutrition Centre developed the Dutch ‘Guidelines for Healthier Canteens’ [6]. These guidelines were developed in collaboration with future users and experts in the field of nutrition and health behaviour and are based on Dutch nutritional guidelines, experiences with the Dutch Healthy School Canteen Program, and research on influencing food choices and nudging [6,7,8,9]. These canteen guidelines aim to assist stakeholders in school, sports, and worksite canteens to create a healthier canteen. According to these guidelines, a healthier canteen increases the offer (availability) and presentation/promotion (accessibility) of healthier products, by using three incremental levels: bronze, silver, and gold [6].

As stakeholders need support to increase compliance with guidelines [10,11], an implementation plan based on their perceived factors that hamper or enable implementation is needed [12]. The implementation plan to support implementation of the Guidelines for Healthier Canteens was developed in collaboration with stakeholders and based on behaviour change models and implementation strategies [13,14,15,16]. Stakeholders gave their input about their experienced or expected barriers or facilitators regarding implementation of school canteen guidelines. The implementation plan aims to address these factors. To evaluate the impact of the implementation plan, changes in these factors should be studied [17]. Such involved barriers or facilitators can arise within the person, as motivation, attitude, and skills or can arise from the environmental context of school or the guidelines, as support from the organisation and the possibility to adjust the guidelines to your own context [18]. To date, the impact of supportive implementation of school based policies on changes in individual or environmental factors is seldom assessed [17].

Studies have shown that implementing school based interventions as intended (fidelity) is a challenge, and that better implementation results in greater effect [19,20]. Insight into the quality of implementation through process evaluation concepts such as fidelity and dose received (completeness) is therefore useful [21,22], as proper evaluation can reveal why an intervention is (not) effective and how it can be optimized [23]. This study evaluated in Dutch secondary school canteens: (a) the effect of the combination of implementation tools on individual and environmental factors affecting implementation as perceived by stakeholders; and (b) the quality of implementation of each tool. 

## 2. Materials and Methods 

### 2.1. Design

We used a quasi-experimental study design involving twenty Dutch secondary schools. Ten intervention schools were asked to implement the recently released ‘Guidelines for Healthier Canteens’ for six months (October 2015 to June 2016). Ten matched control schools received only general information about the ‘Guidelines for Healthier Canteens’. It was aimed to spread intervention schools equally on the main school (canteen) characteristics: catering by a company or by the school itself, schools with below or above and including 1000 students, different levels of secondary education (vocational, senior general, pre-university). To include comparable control schools, control schools were matched on these main and, if possible, on some additional characteristics; availability of shops near the school; and policy for students to stay on the schoolyard during breaks. Sample size calculation showed 20 schools should be included, with 100 students per school. This calculation was based on the effect outcome: students’ purchase behaviour, with a multi-level structure of students within schools (with a correlation of 0.05 between schools), an expected 10% drop-out, 80% power, and 5% significance level. Detailed information about the study design, intervention, and effect evaluation has been described previously [13]. The study protocol was approved by the Medical Ethical Committee of the VU University Amsterdam (Nr. 2015.331) and registered in the Dutch Trial Register (NTR5922).

### 2.2. Study Population

With support of the Netherlands Nutrition Centre and school caterers 155 schools were asked to participate. In total 21 secondary schools (in the Netherlands, schools for students aged between 12 and 18 years) were included. After inclusion, one school dropped out due to organisational problems. The inclusion criteria were: (a) presence of a canteen, (b) intention to make the school canteen healthier, and (c) willingness to provide time and space for the investigators to measure outcomes among students, employees, and canteen workers. The exclusion criteria were: (a) the school had begun implementing the Guidelines for Healthier Canteens and (b) in 2015, the school canteen had received on site support from school canteen advisors of the Netherlands Nutrition Centre. Included schools were located in the central and western part of the Netherlands. All schools received a small financial incentive after completing the study, as notified beforehand.

In all participating schools, the contact (the ‘school coordinator’) identified the stakeholders involved in their school canteen. These were: teachers, representatives of the school board/school canteen or caterer, community health promoters, and students. Due to organisational differences, the number of stakeholders and their function differed per school. Besides, in the intervention schools, the community health promoters wanted to be involved from the start, and in control schools, they wanted to be involved after the research. 

### 2.3. Intervention

We developed an implementation plan by a 3-step approach based on the ‘Grol and Wensing Implementation of Change model’. In short, this model supports a stepwise development of implementation plans by offering six steps, ranging from the development of guidelines to continuous evaluation and improving the implementation process [24]. Our implementation plan consists of several tools aimed to implement the Guidelines for Healthier Canteens in Dutch secondary schools. First, to identify perceived barriers and facilitators to creating a healthier school canteen, semi-structured interviews were conducted among different stakeholders. Second, these factors were prioritized through an expert meeting with 25 attendees from research, policy, and practice. Third, using behaviour change taxonomies and implementation strategies, the factors were translated into implementation tools [12,14,15,16]. This implementation plan was built upon the healthy school canteen program [7]. Table 1 summarizes each intervention tool. A more detailed explanation is available separately [13]. The tools were offered by a school canteen advisor of the Netherlands Nutrition Centre, in collaboration with the Vrije Universiteit Amsterdam. Control schools received only information about the study procedure, the measurements, and general information about the guidelines. After the study, control schools received the same implementation plan as the intervention schools.

### 2.4. Data Collection

Before and after the intervention, school coordinators and the stakeholders completed an online questionnaire about their characteristics and perceived individual and environmental factors affecting implementation based on the validated “Theoretical Domains Framework Questionnaire for Implementation (TDF)” [18] and the “Measurement Instrument for Determinants of Innovations (MIDI)” [25]. The school coordinator was also asked to provide general (organisational) information about the school. After the intervention, the questionnaire for stakeholders of intervention schools was extended with questions based on Saunders (2015) and the MRC [21,23] to evaluate each implementation tool. These answers were discussed in an evaluation meeting. Finally, objective online registered data about the delivery and use of each online tool (schools’ and stakeholders’ questionnaire, the online community, and the newsletter) was collected. For example, for Facebook the amount of sent invitations, registrations, posts, reads were counted. Appendix A provides the questions.

### 2.5. Measures

#### 2.5.1. School Characteristics

Assessed school characteristics were: number of students, education streams at the schools (Vocational/Senior General/Pre-university education), existence of healthy food policy of the school (Yes/No/I do not know), organisation of the canteen (arranged by: external catering organisation/school), presence of vending machines (Yes/No), whether students purchase in the school surroundings such as supermarkets or snack bars (Yes/No/I do not know), presence of a healthier school canteen team or action plan (No/No but intended/Yes). Information about the encouragement of drinking water (Yes/No) and availability of policy for a healthier school canteen (Yes/No) was retrieved from the Canteen Scan (on online tool to assess the availability/accessibility of food and drink products offered in the canteen, see Table 1) completed by the school canteen advisor. 

#### 2.5.2. Factors Affecting Implementation

The implementation plan aimed to change factors which hinder implementation, identified by interviews with stakeholders. These perceived factors that can affect implementation, were assessed by stakeholders with questions derived from the TDF [18] and the MIDI [25]. In accordance with these models, both perceived individual factors, including determinants such as knowledge, self-efficacy, motivation and attitude and perceived environmental factors, including determinants such as need for support, innovation and organisational support, were measured with a five point scale (from 1 = totally disagree, to 5 = totally agree) with 31 and 12 questions respectively. Determinants consisting of more than one item were tested on reliability with Cronbach’s Alpha and analysed separately if lower than *p* < 0.70 [27]. Appendix A provides this information.

#### 2.5.3. Quality of Implementation

To evaluate the quality of each implementation tool, different process evaluation concepts were measured quantitatively [21,23]. Fidelity was measured by dose delivered and dose received. To assess dose delivered, the number of stakeholders provided with the tool by the school canteen advisors or researcher was recorded. Dose received was measured by asking whether stakeholders had received, read, and used the implementation tool. Participant satisfaction with each tool was measured on a 5-point Likert scale (from 1 = totally disagree, 5 = totally agree). Depending on the complexity of the tool, multiple questions were used. Reliability of composite concepts was tested with Cronbach’s Alpha and analysed separately if lower than *p* < 0.70 [27]. Open-ended questions in the stakeholders’ questionnaire and during an evaluation meeting collected additional information: an explanation of the satisfaction score; a short evaluation per tool; an overall evaluation, positive and negative experiences of the total implementation plan; and suggestions for improvements (Appendix A). This qualitative data aimed to clarify the quantitative data.

In addition, objective online registered data about the delivery and use of each online tool were collected. For the questionnaires, the number of sent, started and completed questionnaires were registered automatically. For the online community, Facebook recorded the number of invited and subscribed people and amount of reads per post. For each newsletter, statistics have been recorded of the number of people which have been: (1) sent the newsletter, (2) read it, (3) clicked on a topic to read more. As the online community and the newsletter contained several posts/newsletters, an average has been taken separately for each registered item.

### 2.6. Statistical Analysis

School characteristics were described, and linear mixed model analyses were performed to identify differences in factors affecting implementation (dependent variable) between the intervention and control groups (independent variable). The analyses were done at both stakeholder (level 1) and school level (level 2) by including a random intercept for school in all analyses, because of the assumption that stakeholders within one school were more similar to each other, compared to stakeholders of other schools. We adjusted for the baseline measurement because of any potential differences between groups at baseline. Since the mixed model analyses revealed negligible between schools variance (threshold ICC < 0.20) [28], linear regression analyses were performed. 

For the quality of implementation, mean scores were calculated for each implementation tool, per evaluation concept, and complemented by information collected by open-ended questions. These data were analysed qualitatively by hand using Microsoft Excel, by two researchers independently, following the Thematic Content Approach [29]. First, answers were labelled with objective, descriptive codes; second, codes were split, merged, and interpretative codes were created; third, codes were compared, correlations identified, and overarching themes were formed. Statistical analyses were performed with MLwiN version 2.36 (Centre for Multilevel Modelling, University of Bristol, Bristol, England) and IBM SPSS Statistics version 24.0 (IBM corporation (IBM Nederland), Amsterdam, The Netherlands).

## 3. Results

### 3.1. Characteristics of the Schools

Table 2 provides school characteristics. Most included schools already organised relevant activities (e.g., encouragement of drinking water, availability of policy, a workgroup or action plan). More intervention (*n* = 5) than control (*n* = 2) schools created a policy to restrict students to take unhealthy or big portions of food to school. 

### 3.2. Characteristics of the Stakeholders

A total of 51 stakeholders (27 of intervention and 24 of control schools) started the stakeholders’ questionnaire at baseline. Eleven cases were excluded as they did not fill out the questionnaire at follow-up. Seven cases were excluded due to incomplete questionnaires. In conclusion, thirty-three stakeholders (17 from intervention and 16 from control schools, 1–3 per school) could be enrolled (response rate 64.7%) to analyse the changes in factors affecting implementation. In both the intervention and control group, their roles were: employee at school, as health care coordinator, teacher or facility manager (64.7% vs. 56.3%); employee at a caterer (17.6%, vs. 25.0%); director of a caterer (11.8% vs. 18.8%); or a community health promoter (5.9% vs. 0%). No community health promoters were involved in the control schools, as they wanted to be involved after the research. Some catering companies cater canteens in multiple schools. Stakeholders involved in multiple intervention or control schools (*n* = 4), filled out the questionnaire only once. However, as one catering employee was involved in intervention and control schools, this response was taken into account in both groups, as the experiences were derived from intervention and control schools.

The quality of the implementation tools was assessed by 27 stakeholders in the intervention schools and 7 additional stakeholders: new staff included in the implementation process just after the baseline measurement and after informed consent was obtained. Hence, 24 stakeholders of the 34 that received the implementation tools (response rate 70.6%) evaluated the quality of the implementation tools by completing the quantitative (Table 4) and qualitative questions after the intervention. One to four stakeholders per intervention schools were involved. Their roles were employee at school (62.5%); employee at, or director of a caterer ((12.5% resp. 16.7%); or a community health promoter (8.3%).

### 3.3. Factors Affecting Implementation

Table 3 shows, at follow-up (T1), compared to the control schools, the intervention schools scored higher on the factor knowledge (only ‘I have all the information I need, to make the school canteen healthier’) and motivation and lower on need for support. The determinants descriptive norm and perceived organisational support showed marginal differences between intervention and control schools after intervention.

### 3.4. Quantitative Evaluation of the Quality of Implementation

Each implementation tool was delivered in every intervention school. As planned, three tools were delivered only to the school coordinators, the others to all involved stakeholders. The advisory meeting was adapted based on their results of the schools’ and stakeholders’ questionnaire and the Canteen Scan. The students’ factsheet was also school specific, based on their own students’ answers. Table 4 shows that a majority of stakeholders indicated attending/receiving and reading the advisory meeting and report (67.9% and 64.3%, respectively), the communication materials (60.7% and 50.0%) and the fact sheet (80% and 60%). According to the objective collected data, more stakeholders subscribed to or read the online community and the newsletter (61.8% and 45.0%, respectively). For the online community, this number is higher than measured with the questionnaires (21.4%). The implementation tools, (i) advisory meeting and report, (ii) communication materials, and (iii) fact sheet, had the highest mean (SD) scores on satisfaction, 4.17 (0.44), 3.98 (0.23), and 4.31 (0.40), respectively. 

### 3.5. Qualitative Evaluation of the Quality of Implementation

The questionnaires for schools and stakeholders were evaluated as being too long and some questions as being difficult to answer if the participant had limited involvement in canteen activities (e.g., school directors or community health promoters). While, due to technical limitations, some participants did not fill out the Canteen Scan themselves, all received the result of the Scan filled out by the school canteen advisor. The Scan was rated as added value increasing knowledge, providing insight into and monitoring the level of healthiness of the canteen over time. Stakeholders were satisfied with the personal contact with the school canteen advisors, insight received into their canteens and the tailored, clear, and feasible advices. Stakeholders of schools and caterers both mentioned the importance of collaboration with each other, knowing each other’s expectations, and defining aims and actions together. The advisory meeting helped strengthen this. 

Stakeholders evaluated the communication materials as clear, feasible and inspiring. The newsletter was also evaluated as feasible and useful, especially as a reminder, for inspiration, and for tips. The newsletter as information overload, in combination with other health related newsletters stakeholders received, was mentioned. Sharing online information, advice, and news by the online community was evaluated positively while time constraints and Facebook as chosen medium were mentioned as limitations.

Due to potential privacy sensitivity, the students’ wishes and needs fact sheet was sent only to the school coordinator who could choose to share it with other stakeholders. Some stakeholders were dissatisfied not receiving it, indicating that the fact sheet was not shared. Stakeholders evaluated the fact sheet as a positive method to get student’ opinion and the support of colleagues. One limitation was that the fact sheet was based only on second grade students.

Overall, stakeholders mentioned the combination of different implementation tools as positive. They used the tools they considered appropriate to their situation and preferences. They mentioned several preconditions to realizing a healthier school canteen: sufficient time, money, and facilities; freedom at the work place to perform activities related to a healthier school canteen; adequate knowledge and examples about healthier products and accessibility; existence of a multidisciplinary workgroup; clear and timely information about the guidelines, including possible future changes; the possibility of involving students; and sufficient customers.

Challenges mentioned include, first, lack of support from the school’s neighbourhood due to the existence of numerous selling points and offers of less healthy products. Second, competing demands related to other school tasks, such as educational tests, rebuilding or staffing problems, make keeping the healthier school canteen on the agenda difficult. Third, involving students and colleagues and alignment of all health-related activities in the school was found to be important but challenging. Fourth, while many stakeholders learned that the accessibility criteria lead to behavioural changes in students, some did not understand how, or which criteria could be used. Fifth, although stakeholders experienced inconsistency in the financial effects of a healthier school canteen (some schools noticed lower and others higher sales) and long-term effects are unclear, they were wary of potential negative financial consequences.

## 4. Discussion

This study evaluated the process of implementation of the Guidelines for Healthier Canteens in secondary schools in the Netherlands. First, it showed that implementation support resulted in changes in individual and environmental factors related to the implementation of healthier school canteen. More specific, knowledge, descriptive norm, motivation, and perceived organisational support increased, and need for support of stakeholders decreased. Second, stakeholders evaluated the implementation tools positively, especially the advisory meeting and report, the students’ fact sheet, the communication materials, and the ‘Canteen Scan’. 

The implementation plan improved both some individual and environmental factors, although changes are small. However, these changes are supported by the qualitative results. Stakeholders indicated that the plan supported them in creating a healthier canteen. Their positive feeling of support and increased knowledge and motivation may lead to better implementation [30]. Only a few other studies evaluated the process of supportive implementation of school health policy, and they showed mixed effects on individual factors, such as the relation between being interested and (not) implementing a health related school based intervention [17].

The results with regard to the second research question showed that the personalization and combination of tools particularly supported stakeholders in the implementation of healthier canteen guidelines. Stakeholders considered it helpful to receive personal advice and to use the tool suitable to their specific situation. For example, during the advisory meeting, the given personal advice was helpful to draft aims, supported by stakeholders of school and caterer. Hence, the newsletter reminded them to remain active and to keep the canteen on the agenda. In addition, the students’ opinions, summarized in the factsheet, supported stakeholders to discuss the healthy canteen topic with colleagues. These results are in line with Australian studies showing that implementation of healthy canteen policies can be achieved in most schools with multi-strategic support, including personalized support, monitoring, and feedback [31,32]. 

Although the satisfaction with the advisory meeting and communication materials could be influenced by their high use [33], the qualitative results also indicated that the personal contact, tailored advise, examples of healthier products/accessibility, and information about the guidelines given by these tools inspired them. In contrast, the online community scored low on satisfaction. The choice to use Facebook as medium could have influenced these results. Stakeholders indicated they only wanted to use Facebook in personal life. In addition, a supportive community will only be reached if enough people actively contribute. Due to the research setting in which only a limited number of schools and stakeholders participated, we were only able to set up a limited community. The number of subscribed stakeholders (*n* = 21) may be too few to realize an active community. Outside the research setting, more people could subscribe and interact, which may result in higher use and support. 

In contrast to the high use of the advisory meeting and communication materials, only four stakeholders used the Canteen Scan themselves. This could be explained by the delayed development of the tool, which made it difficult for stakeholders to fill out the scan by themselves. However, in all schools, school canteen advisors filled out the Canteen Scan and discussed the results in the advisory meeting. Stakeholders indicated that insight into the level of their canteen and tailored advices to improve the canteen as generated by Canteen scan helped them to define aims and actions. Our results agree with earlier research, in which tailoring programs to schools’ needs and context, ownership, and providing support and examples were found to be effective to implement school based interventions [34]. These findings could be explained by the different characteristics and diverse and dynamic social, physical, and organisational context of schools and their canteens, which make general advice less applicable. 

### 4.1. Strengths and Limitations

A strength of this study is the involvement of stakeholders from schools, community health services, caterers, and the Netherlands Nutrition Centre during the process of development and evaluation. This enabled identification of a wide range of factors affecting implementation from different perspectives. We were therefore able to develop tools that were broadly supported, engaged different stakeholders, and could be easily integrated into existing school routines. We evaluated the tools using a combination of qualitative and quantitative data collected through questionnaires, an evaluation meeting, and online registered data. This combination resulted in reliable and broad insight into both the effects of the tools on perceived factors affecting implementation and the quality of implementation and also provided indications for improvement.

The limitations of this work include first, that we only had data from twenty schools and a relatively small number of stakeholders per school. Included stakeholders, like representatives of caterers and school canteen advisors, represent or visit a large number of schools, thus extending the range of schools affected. Within our study, four caterer employees were involved in multiple schools, of which one was involved in intervention and control schools. This could have biased the results as the received intervention could have influenced the control schools. It is possible that this made the differences between intervention and control schools smaller. Hence, as it was only one person, the bias will be negligible. Second, as mentioned, the Canteen Scan was still in development during data collection. Consequently, school canteen advisors of the Dutch Nutrition Centre could fill in the scan, but for many stakeholders, this was still too difficult. This resulted in low uptake. Stakeholders responded positively to the score and advice generated by the Canteen Scan after being filled out by school canteen advisors. This resulted in the Canteen Scan being improved after this research study. Reasonably, this would improve the use for stakeholders. Third, as all included schools were already motivated to implement the guidelines, stakeholders may have been more positive about their perceived individual and environmental factors regarding implementation of school canteen guidelines than non-included schools. This may have resulted in an underestimation of the tools’ effect. Finally, as fidelity is an important concept to measure the quality of implementation [21,22,23], we measured it using a combination of dose delivered, dose received, and satisfaction. However, previous studies show that measuring fidelity in multi-component, tailored interventions is difficult and yet, there is no consensus about how to measure it [22]. To be able to compare the quality of implementation across studies, it is recommended to clearly define and use one consistent method to assess fidelity and other process evaluation concepts [22]. 

### 4.2. Implications

As also recognized in a previous study [35], creating support and involvement of students, colleagues, and stakeholders within and outside the school and keeping the healthier school canteen on the agenda are both essential and a challenge. Regular reminders such as newsletters, regular contacts with the school canteen advisors, and prompts to fill in the Canteen Scan helped schools to continue paying attention to a healthier food environment. To support sustainable implementation, a healthier school canteen should be aligned with other school health policy, combined with environmental policy to influence the surroundings of the school. To keep stakeholders involved, regular monitoring and feedback of the food environment by measuring the availability and accessibility of healthier food and drink products in canteens and also of students’ wishes and needs are recommended. However, in the Netherlands, schools are not obliged to offer and promote healthier foods or drinks at schools. For this reason, our implementation plan will only support schools that voluntarily want to take action.

To further improve the implementation plan and continue national implementation of the ‘Guidelines for Healthier Canteens’ in Dutch secondary schools, our results and learnings were shared with the Netherlands Nutrition Centre. Based on these results, implementation tools were improved. For example, the Canteen Scan was improved by adding more explanations and an explanation video ‘how to use the scan’ was created. Moreover, regarding the fact sheet of students’ needs and wishes, we recommended schools to use input of students of different educational levels and grades.

The guidelines for healthier canteens are applicable to sports canteens as well. For this reason, the insights were also shared with stakeholders involved in creating healthier sports canteens. Further research is needed to show whether the findings in the present study are applicable to other settings (such as sports canteens and worksite cafeteria’s), other countries, and other health related school based interventions. Moreover, further research is needed to gain more insight into processes of implementation and to be able to compare the quality of implementation across studies. In our opinion, comparability could be improved by clear definitions of concepts like fidelity, dose received, and dose delivered, as well as clear operationalizations to measure these concepts [22,36]. However, this is challenging because it is also recommended that these measures be adaptable to implementation tools in a specific context. 

## 5. Conclusions

In conclusion, the tools to implement the Guidelines for Healthier Canteens seem to result in positive changes with regard to individual and environmental factors affecting implementation. The combination of implementation tools supports stakeholders in creating a healthier canteen. In particular, the tools that included students’ wishes, tailored information and feedback, reminders, and examples of healthier products/accessibility were evaluated positively.

## Figures and Tables

**Table 1 ijerph-16-04509-t001:** Description of the implementation plan to implement the Guidelines for Healthier Canteens ^a^.

Implementation Tool	Action and Targets	Target Group	Period
1. Insight into the current situation			
*1.1. Questionnaire, school*	The results of the online questionnaire to assess and provide insight into the characteristics of the school [18,25].	Coordinator of the school	Before the advisory meeting
*1.2. Questionnaire, stakeholders*	The results of the online questionnaire to assess and provide insight into stakeholders’ characteristics, individual and environmental determinants [18,25].	All involved stakeholders	Before the advisory meeting
*1.3. ‘Canteen Scan’*	An online tool that provides (I) insight into and (II) directions for improvement of availability and accessibility of food and drink products in canteens [26]. All available products can be entered, the tool will automatically classify product in healthier/less healthy product, according to the Dutch nutritional guidelines. Closed questions assess the accessibility, availability of water, and presence of policy. To create ownership and insight into the changes so far, the school receives information to fill out the Canteen Scan by themselves if they wanted.	Performed by a school canteen advisor of the Netherlands Nutrition Centre and by the school coordinator. Results and feedback provided to all involved stakeholders.	Before the advisory meeting (by the advisor) After three months (by the coordinator)
*1.4. Advisory meeting and report*	In one advisory meeting per school, all involved stakeholders are advised about how to improve the canteen by a school canteen advisor of the Netherlands Nutrition Centre. Based on the aims of the school and the points of attention, identified with the two questionnaires and the Canteen Scan a concrete action plan will be developed. This action plan is created together to increase ownership and collaboration. After the meeting, a written report based on this meeting is distributed by email.	All involved stakeholders	At the start of implementation
2. Communication materials	A brochure about the Guidelines for Healthier Canteens, an overview of the steps to take, a personalized poster, a banner for the schools’ website. To create motivation and increase and apply knowledge. Content: information, examples of healthier products, how to place products, and healthier canteens.	Coordinator of the school, who is asked to share this with other stakeholders.	At the start and halfway through implementation
3. Online community	A closed Facebook community for stakeholders was established to share their experiences, ask questions and support each other.	All stakeholders	Continuous
4. Digital newsletter	A regular newsletter sent by email, consisting of information and examples regarding the healthy school canteen. To support, remind and motivate stakeholders.	All stakeholders	Once every 6 weeks (4 in total).
5. Students’ fact sheet	A summary of their students’ wishes and needs with regard to a healthier school canteen, to gain insight into the opinions of students and how students want to be involved.	Coordinator of the school, who is asked to share this with other stakeholders.	Once, 2–4 weeks after the start.

^a^ This table is adapted from the version published in the design paper [13].

**Table 2 ijerph-16-04509-t002:** Characteristics of the participating intervention and control schools.

Characteristics of the Schools	Intervention Schools (*n* = 10)	Control Schools (*n* = 10)
*Amount of students*		
Mean (SD)	928 (509)	1145 (503)
Range	215–1926	330–1720
*Educational level (n)*		
Only vocational	3	3
Only senior general/pre-university	2	3
Vocational and senior-general/pre-university	5	4
*School canteen catering (n)*		
Arranged by:		
Caterer	7	8
The school	3	2
Offered via:		
Only counter	0	1
Counter and vending machine	10	9
*Basic Conditions Healthier Canteens (n)*		
Encouragement to drink water (Yes)	5	6
Policy available for a healthier school canteen (Yes)	1	1
*Organised regarding school canteen (n)*		
Workgroup		
No	1	4
No but intention	3	2
Yes	6	4
Action plan		
No	1	3
No but intention	5	2
Yes	4	5
*Available school policy (n)*		
Policy to stay at the schoolyard, Yes	9	8
Policy which forbids to take certain foods to school (like big portions, energy drinks)		
Yes	5	2
No	4	7
I do not know	1	1

**Table 3 ijerph-16-04509-t003:** The factors affecting implementation perceived by stakeholders and differences between intervention and control at follow-up (T1).

Factor Mean (SD)	Intervention (*n* = 17)		Control (*n* = 16)		Linear Regression Analyses	
	T0	T1	T0	T1	Beta	CI
*Individual Factors*						
Knowledge						
Role clarity: Clear what activities to do to make the school canteen healthier	3.94 (0.83)	4.29 (0.77)	3.69 (1.14)	4.06 (0.85)	0.22 (0.29)	−0.37; 0.81
Knowledge: I have all the information I need to make the school canteen healthier	3.29 (1.11)	4.24 (0.75)	3.38 (1.03)	3.63 (0.96)	0.61 (0.30) *	0.00; 1.23
Knowledge: I have enough knowledge to make school canteen healthier	3.94 (0.83)	4.18 (0.53)	4.06 (0.77)	3.94 (0.68)	0.27 (0.21)	−0.16; 0.69
Self−Efficacy	3.59 (0.54)	3.34 (0.76)	3.68 (0.92)	3.71 (0.85)	−0.02 (0.25)	−0.53; 0.48
Attitude	3.78 (0.56)	4.03 (0.50)	3.72 (0.89)	3.81 (0.41)	0.21 (0.15)	−0.10; 0.52
Social influence						
Descriptive norm: Colleagues perform their healthier school canteen activities good	2.82 (1.55)	4.00 (0.79)	3.56 (0.63)	3.62 (0.96)	0.60 (0.30)	−0.08; 1.20
Subjective norm: Other people expect me to perform my healthier school canteen activities good	3.82 (1.13)	3.88 (1.54)	4.00 (0.73)	3.81 (0.83)	0.20 (0.36)	−0.53; 0.94
Social support: I receive enough support in performing my healthier school canteen activities	3.41 (1.18)	3.71 (1.16)	3.75 (0.93)	3.69 (1.08)	0.13 (0.38)	−0.65; 0.91
Routine	3.09 (1.28)	3.47 (1.14)	3.44 (1.11)	3.38 (0.79)	0.15 (0.35)	−0.55; 0.86
Intention	3.76 (1.14)	4.12 (1.32)	4.38 (0.81)	3.88 (1.50)	0.25 (0.52)	−0.82; 1.32
Motivation	4.41 (0.51)	4.65 (0.49)	4.38 (1.26)	4.19 (0.66)	0.45 (0.20) *	0.05; 0.86
Skills	3.82 (1.13)	4.29 (0.47)	4.00 (1.21)	4.12 (0.62)	0.17 (0.19)	−0.22; 0.57
Professional Role	3.76 (1.12)	4.12 (1.27)	4.00 (0.88)	3.94 (0.87)	0.37 (0.26)	−0.15; 0.89
Behavioural regulation	3.08 (0.79)	3.53 (0.64)	2.88 (1.13)	3.38 (1.17)	0.06 (0.29)	−0.54; 0.66
*Environmental factors*						
Need for support	3.47 (1.05)	2.61 (0.79)	3.10 (0.99)	3.29 (0.90)	−0.79 (0.29) *	−1.37; −0.21
Innovation						
Consistent with my usual work	3.88 (0.93)	3.71 (1.45)	3.94 (0.93)	4.06 (1.00)	−0.36 (0.44)	−1.26; 0.54
Adaptable to the vision of school	3.82 (1.19)	4.06 (1.44)	3.75 (0.86)	3.75 (0.93)	0.25 (0.29)	−0.34; 0.83
Perceived organisational support	3.33 (0.68)	3.54 (0.46)	3.36 (0.65)	3.21 (0.79)	0.35 (0.19)	−0.04; 0.74

* Significant differences between intervention and control group after the intervention tested with linear regression model, corrected for baseline measurement, *p* < 0.05.

**Table 4 ijerph-16-04509-t004:** Quality of implementation per implementation tool.

Implementation Tool	Target Group	Dose Delivered and Received Objective *n* (%)		Dose Received Subjective ^a^ *n* (%)		Satisfaction ^b^ Mean (SD)
Questionnaire school	Each school	Invited	10	-		3.56 (0.88)
		Started	9 (90.0%)			
		Completed	9 (90.0%)			
Questionnaire stakeholder	All stakeholders	Invited	46	-		3.40 (0.87)
		Started	34 (73.9%)			
		Completed	24 (52.2%)			
Canteen Scan	Each school	Invited	10	Used	3 (30%)	3.50 (0.66)
Advisory meeting and report	All stakeholders	Sent to	27	Received	19 (67.9%)	4.17 (0.44)
				Read	18 (64.3%)	
Communication materials	All stakeholders	Given to the stakeholders present at the meeting		Received	17 (60.7%)	3.98 (0.23)
				Read	14 (50.0%)	
Online community	All stakeholders	Invited	34	Subscribed	5 (17.86%)	2.61 (1.31)
		Subscribed	21 (61.8%)			
		Read	17 (50.0%)			
Newsletter (was sent 4 times)	All stakeholders	Sent to	34	Received	13 (46.4%)	3.35 (0.58)
		Average read	15.3 (45.0%) Range per newsletter 14–17	Read	9 (32.1%)	
		Average click on topic	4.8 (14.1%) Range per newsletter 2–6			
Students’ fact sheet	Each school	Sent to	10	Received	8 (80%)	4.31 (0.40)
				Read	6 (60%)	

^a^ Dose received was measured by 1, 3, 5, or 6 questions, with a 5-point Likert scale (1 = totally disagree, 5 = totally agree). To calculate the percentage, the 24 persons who filled in the questionnaire were taken as 100%, except for the Canteen Scan and Students’ fact sheet were 10 persons who received these materials are 100%. ^b^ The questions to assess Satisfaction were answered by the stakeholders who used/read/completed the implementation tool. Satisfaction was measured by 1 to 6 questions, depending the implementation tool (see Appendix A), with a 5-point Likert scale (1 = totally disagree, 5 = totally agree).

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
