# Peer review of "Implementation of Guidelines for Healthier Canteens in Dutch Secondary Schools: A Process Evaluation"

_ijerph, 2019, doi:10.3390/ijerph16224509_

Round 1
Reviewer 1 Report
The paper presents the approach and findings from the process evaluation of a real-world implementation of healthier canteen guidelines. I applaud the study team for having built in a process evaluation, which often times is forgotten in such real-world studies. However, I think the manuscript could benefit from providing more details and having a deeper interpretation of the findings, and richer discussion of the implications from this process evaluation effort. Please find below major points of concern, and minor points on typos, formatting, and expression matters.
Major Points
Lines 172 to 175:
The number of stakeholders and the breakdown is confusing as there seems to be a mix of those that filled in only the baseline questionnaire with and without the follow-up questionnaire, and those that filled in only the follow-up questionnaire. Additional information should be provided on why a scenario occurred whereby there were some staff that completed only a follow-up questionnaire. Were these new staff that have yet to join at baseline, or had somehow missed or opted not to complete the baseline questionnaire but somehow completed the follow-up questionnaire. If so, were they properly consented to be enrolled in this study before completing the questionnaire?
Since the stakeholders are the key participants providing data used in this paper, there should first be a section clearly outlining the numbers for stakeholders that received the intervention materials, completed the “factors affecting implementation” questionnaire, and completed the “quality of implementation” questionnaire.
Line 185 (Table 3):
If there were 5 stakeholders that were involved in multiple schools, this fact should be presented in the main text and not just in the footnote for Table 3. Additionally, it is unclear how the authors were then able to account for this in their ICC calculations in lines 155 to 158.
Lines 162 to 164 and Lines 211 to 248:
Was any thematic analyses software used?
How many stakeholders completed the open-ended portion of the questionnaires? How many different schools were they from?
It is unclear if the reported qualitative findings could be considered to be representative of the sample and the intervention processes, and if there was sufficient unprompted open-ended data to reach saturation.
Table 3:
It is unclear what the reason is for the 4th heading of “Linear regression analyses” to have the term “crude”. This should be explained in the statistical analyses section (or better linked if it’s already explained in there now).
Table 4 “Newsletter”:
The number presented for dose of newsletters read and clicked on has a decimal, which indicates this is likely a mean number generated. It is, however, unclear what this is a mean of, what the denominator is, and how it was calculated. Since many newsletters were sent over the course of the intervention, the team should either present the mean percentage of how many newsletters they read or clicked on within each stakeholder (i.e. if Stakeholder 1 received 4 and read 3 it is 75%, and if Stakeholder 2 received 4 and read 1 it is 25%, and the mean of these two is 50% read), or present the count and proportion of stakeholders that read or clicked on at least 1 (i.e. if using previous example, both stakeholders read at least 1, and so it is a 100% that read at least 1).
Lines 256 to 269:
This paragraph reads too much like the strength of this study, with a few sentences summarizing some results. The point of the front part of the discussion section should present key findings and how to interpret them together, particularly for a process evaluation with so many aspects of assessments that cover both quantitative and qualitative parts. At current state, this section reads a little too vague and cursory, without providing more meaningful insights.
Lines 270 to 275:
This is a summary of the qualitative findings. The authors need to go more in depth in the discussion section and take it to the next level to properly showcase the findings of both the qualitative and quantitative sections of this study.
Take the example of the Canteen Scan, the reality was that the uptake of the tool was relatively low with 4 out of 10, and so the positive ratings come only from those that would have used it. The authors have to explain the situation more of what the underlying reasons were for this low uptake and what that means. It was mentioned in the results section that there were “technical limitations”, which gathering from the limitations section, that this was perhaps due to delayed development and launch of the tool for use. So was this a sufficient explanation for why there was low uptake of this tool? Does more work need to be done to refine the ease of use of the tool for stakeholders without sufficient time?
This section also failed to discuss more in depth the findings for the online community. The self-reported subscription to the Facebook group was only 6 out of 38, and the reported satisfaction was a mean of 2.63, which is low compared to the other satisfactions scores with means 3.40 and above. There was a line in the qualitative section of the results mentioning that Facebook as a medium was a limitation. The authors should discuss this more and give their guess on why this was the case of the low rating and mentioning of Facebook as a limitation, and provide their views on how this tool could be improved, or if it should be eliminated if minimal in its contribution of support to the guidelines implementation.
Lines 297 to 299:
The sentence should be rephrased to more explicitly point out the limitation. Is the limitation in the fact that the Canteen Scan was delayed in being developed and launched and so affected the uptake? Is the limitation in the assessment of Canteen Scan being limited due to the low uptake and so a very small sample?
Lines 318 to 321:
It is unclear how these results were used to improve the implementation plan and in what areas. The actual recommendations derived from this study that would have been shared with the Netherlands Nutrition Centre should be further described. For example, for the package of tools, some areas in need of improvements were brought up by the stakeholders’ responses: Canteen Scan (technical issues and low uptake?), newsletter (too much information), and fact sheets (not being circulated and only on 2nd graders). Were these tools then further fine-tuned? Were more tools added to the package for schools to pick from to suit their needs?
Minor Points
Table 1 for “4. Digital newsletter”:
“A regularly newsletter…” should be “A regular newsletter…”
“Every 6-week.” would be better as “Once every 6 weeks.” Additional detail of how many total newsletters should be stated as well without the reader having to calculate that it was 4 newsletters based on assumption of 24 weeks of intervention divided by 6 weeks.
Table 2:
Formatting of the table headings and the indentation of the options in the first column could be improved to facilitate the readability.
“Mean (SD)” should also be in the first row of headings and not be in the “Individual factors” line in the “Factor” column.
The n number can be in the “Intervention” and “Control” heading since analyses was only on data from stakeholders with both T0 and T1 responses.
Line 186 (Table 2):
“Significance differences…” should be “Significant differences…”
Line 191:
“This numbers from the number…” should probably be “This differs from the number…”
Table 4 “Online community”:
The count of 17 for “Read” should not have a decimal point since the count can only be a whole number, and to stay consistent with the formatting for “Subscribed”.
Line 230:
The choice of word “regretted” should only be used if the stakeholders that did not receive the fact sheet because they actively opted out and thus regret that decision. If the stakeholders did not receive it because school coordinator did not include them, then these stakeholders did not have a choice in that decision and are dissatisfied with the approach of their colleagues and/or the program implementers.
Reviewer 2 Report
Thank you very much for the opportunity to review this article. This is a relevant study that aims to evaluate how a diversity of tools that may help stakeholders on the implementation of guidelines for healthier canteens. It seems that the authors did a comprehensive research work, however the reporting and the information provided in the manuscript does not show that. I would advise the authors to offer more details about the tools used and their applications. I would also like to see more details on their methodological decisions, such as the statistical analysis. Additional comments are as follows:
Abstract:
Lines 10 – 11: Please give examples of the implementation tools you are citing. A comma after “To facilitate implementation of these guidelines” would help comprehension.
Line 11: please add the study design.
Line 16: and the establishment of an online community…? It seems there is information missing in this sentence.
Introduction
Please add some context on how the school canteens work in the Netherlands. It would be useful for the readers to know if students mostly bring their food from home, or buy it at school? Does the school provide meals based on a national/regional school meal plan?
Line 32: Please add a context on what type of policies you are referencing to. Are you talking about school-based, regional or national policies? Perhaps give examples?
Lines 43 – 56: It is difficult to follow these sentences. Perhaps you can start by saying that you built an implementation plan based on knowledge from previous research and literature. These insights include, identified barriers and facilitators experienced, or expected by different stakeholders, behaviour change models, implementation strategies, and collaboration with stakeholders.
Lines 46 – 48: those sentences are repeating the content of the first sentence on this paragraph.
Lines 48 – 52: Also difficult to follow those sentences. Are the authors trying to say that there are both individual and contextual factors that may affect the implementation of the interventions? Please be clear about it. The authors could list what are those individual (motivation, attitude and skills) and contextual (institutional support, adaptability of the guidelines (?) factors. Please also clarify what you mean by adaptability of the guidelines.
Line 51: what changes? What factors? Here you are trying to state your research gap. Please be as clear as possible.
Line 55: Please introduce and explain these concepts earlier (i.e. process evaluation concepts such as fidelity and dose).
Here and throughout the manuscript, please be more specific on which environment you are focusing on. I understand that your focus is on the school food environment, but readers may be confused with the terminology and also think about the neighbourhood or social food environment. Please clarify the concepts you are using in the introduction or in your methods section.
Methods
Lines 63 – 64: It is nice that the authors may refer to a previously published study to provide detailed methodological decisions. However, a more elaborate, yet brief description on study design and intervention is needed. I would like to see here information such as:
- how many schools where invited?
- Did the authors do any sample calculation to estimate the number of schools needed?
- How was the school sample selected? Randomly, per convenience? Some explanation on that is needed as it may affect generalizability of findings to other schools.
- Did the authors try to balance the sampled schools profile? For instance in terms of region and SEP?
Lines 75 – 76: So the Guidelines for Healthier Canteens was already available? And were there schools already implementing it?
Lines 78 – 79: Did the schools know they would receive a financial incentive?
Line 87: Please briefly explain what this model is about.
Lines 104 – 111: Could you provide more information about this questionnaire? How was it formulated? Was it validate? I would also like to see more information on this online data collection about the delivery and use of each tool.
Line 121: Could the authors share more information about the Canteen Scan? For instance, what type of questions were available?
Lines 153 – 158: Could you please explain in more detail the testing for a multilevel structure of your data? Did you add a random intercept for school (n=20) in your models? Or did you test it only among the intervention schools? Please add your rationale for testing a potential dependency of your data at the school level. Why did you expect that?
Line 158: Statistical analysis: I would like to have more detailed explanation on how the variables were treated in your models. For instance, what exactly was your outcome and exposure measures and, very importantly, how did you operationalize it? Why did you chose to average your ordinal data? And why did you chose a linear model? Are the residuals of your dependent and independent variable normally distributed? Finally, I have doubts that the linear model is appropriate to analyse averaged ordinal data such this from a Likert-scale. It may be better to use a non-parametric test. Perhaps you may want to consult a statistician?
Line 160: typo “these data” instead of “this data”.
Results and discussion
Table 1: what do the authors think about the fact that more intervention than control schools have a policy to forbid certain foods to be taken to school? May it be a source of bias?
Line 79 - 182: Please describe your findings in a more detailed manner. I don’t think that a linear regression is the most appropriate choice for this type of data, but try to think what would it be the interpretation of a significant beta coefficient of 0.59 (0.27)?
Line 186: “significant differences” instead of “significance differences”.
Line 191: This sentence is incorrect, please revise.
Line 240: do stakeholders have opinions on whether or not a zoning ban on unhealthy food retailers around school would be beneficial to promote healthier diets among student? Such a ban is already in place in some cities around the world, that could be mentioned in the discussion.
Line 247: what are these inconsistencies experienced by stakeholders?
Line 251: a verb is missing here.
Line 264 – 165: Are those environmental factors on Table 3?
Generally in the discussion, the authors could indicate how, or if, their implementation plan can be applied nationality. Do they think there is an general interest from other schools to implement the guidelines for healthier canteens in more schools in the Netherlands? Would the implementation of the guidelines be possible to be part of central national plan? Or would it most likely be a choice for each school to implement the guidelines or not?
325 – 326: Concepts such as fidelity, as well as the issues regarding how to assess it, should be introduced earlier in the manuscript. Not much attention is given to these concepts (i.e. fidelity and dose) in the manuscript’s present form.
313: As I mentioned before, please be more specific on which environment you are focusing on.
Reviewer 3 Report
This is an interesting paper that is worth of publication, I only have one question to the authors:
The authors mentioned in Line 81-82: the stakeholders were teachers, representatives of the school board/school canteen or caterer, community health promoters and students. Yet, only 28 stakeholders evaluated the implementation tools (Line 190). The manuscript did not provide the information of how many teachers, representatives of the school board/school canteen or caterer, community health promoters and students in these 28 stakeholders, respectively. This is important, because if the 28 stakeholders were from only one or two type of them (for example, only from students), the evaluation of the tools would be biased, and cannot represent the real situation.
Reviewer 4 Report
This paper investigates the factors affecting the implementation of guidelines for healthier canteens, and compares the quality of implementation using different tools. I have a few comments on the data processing and presenting of results.
What is the definition of healthier canteen? Please clarify. How many implementation tools are employed in your study? Those you mentioned in the abstract is not the same as those listed in Table 1. Please confirm all tools used in your experiment and give the definition of each tool. As ten schools are selected as intervention schools, do all intervention schools receive the same implementation plans? I suggest using some figures to compare the results between control school and intervention school. There are too many words in some tables, which make tables difficult to understand. I think the paper should concentrate on the key aims of the study. That is to say, you can only present the results of your research objective and remove results of others to the supplementary material.
Round 2
Reviewer 1 Report
Thank you to the authors for making the edits and additions. Please find below some major points of concern. There are also quite a few more minor points on typos, formatting, and expression matters, and they all seem to occur in the newly added text. Understandably, a revision timeline is tight, but I implore all authors to all give a close read and contribute in editing the manuscript.
Major Points
Line 223 (Table 3):
In the authors’ response to a previous comment on the footnote in Table 3, the authors have clarified that there are stakeholders involved in multiple intervention or control schools. Additionally, it is now highlighted in this new footnote that one stakeholder was involved in both intervention and control schools. All these details of the study population need to be expanded and addressed in the text in detail, and not just glossed over in a footnote with limited explanation.
This is a limitation of concern and puts in question how the authors dealt with the impact on data and interpretation:
- Do stakeholders involved in multiple schools complete the questionnaires multiple times, each time for each school?
- What does it mean that the authors included the stakeholders involved in multiple intervention/control schools once? Was a mean of their responses taken?
- How reliable can we trust the stakeholders’ ability to separate their responses for each school?
- If there is a stakeholder involved in both intervention and control schools, doesn’t it mean that the control school has now been “contaminated” since the stakeholder has the knowledge and resources gained from the intervention school to be applied to the control school as well?
If the authors cannot fully justify the analysis approach and decisions, then perhaps a reanalysis of the sample of stakeholders should be done. If a reanalysis is not done, then perhaps the claims and conclusions will need to be less strong and with the caveat of limitations and assumptions clearly highlighted. Currently, there seems to be a lot of assumptions and decisions that the authors have had to made in conducting this process evaluation – this is normal for such a real-world evaluation, and the thinking and logic behind these decisions and assumptions should be shared openly and not concealed as these are exactly the valuable learnings that such a paper bring to the table.
Table 4 “Newsletter”:
I now get that the number presented in the table is a mean of the stakeholder readership per newsletter issue. However, this was not a metric that was immediately intuitive, since it was a mean over 4 newsletter issues, and with a further divide of sent to, read, and clicked subdomains of this metric. It would be ideal if the explanations and calculations are described in the text briefly.
Minor Points
Line 36:
The word “were” should be “where”.
Line 72:
The phrase “schools less and above 1000 students” should be “schools with below and above 1000 students”.
Line 160:
The phrase “Depending the” is missing the word “on”.
Lines 316 and 322:
The use of “In contradiction” is awkward, and likely the authors mean “In contrast”.
Lines 319-320:
This sentence is currently a little incoherent and should be rephrased: “only stakeholders subscribed this may be too few to realize an active community.”
Line 344:
The phrase “relatively a small number of” should be “a relatively small number of”.
Line 351:
The sentence “This resulted that the Canteen Scan was improved after the research” should be rephrased to “This resulted in the Canteen Scan being improved after this research study.”
Lines 377-380, 386-388:
The sentences need to be revised. Very awkward in current state.
Reviewer 4 Report
Authors have well addressed all my comments. I have no further questions. I think the manuscript can be accepted and published in IJERPH.
Author Response
No response needed. Thank you very much for your time and effort.